# Effects of Cardiac Stem Cell on Postinfarction Arrhythmogenic Substrate

**DOI:** 10.3390/ijms232416211

**Published:** 2022-12-19

**Authors:** Ángel Arenal, Gonzalo R. Ríos-Muñoz, Alejandro Carta-Bergaz, Pablo M. Ruiz-Hernández, Esther Pérez-David, Verónica Crisóstomo, Gerard Loughlin, Ricardo Sanz-Ruiz, Javier Fernández-Portales, Alejandra Acosta, Claudia Báez-Díaz, Virginia Blanco-Blázquez, María J. Ledesma-Carbayo, Miriam Pareja, María E. Fernández-Santos, Francisco M. Sánchez-Margallo, Javier G. Casado, Francisco Fernández-Avilés

**Affiliations:** 1Department of Cardiology, Instituto de Investigación Sanitaria Gregorio Marañón (IiSGM), Hospital General Universitario Gregorio Marañón, 28007 Madrid, Spain; 2Center for Biomedical Research in Cardiovascular Disease Network (CIBERCV), 28029 Madrid, Spain; 3Facultad de Medicina, Universidad Complutense de Madrid, 28040 Madrid, Spain; 4BSEL—Biomedical Sciences and Engineering Laboratory, Bioengineering Department, Universidad Carlos III de Madrid, 28911 Madrid, Spain; 5Centro de Cirugía de Mínima Invasión Jesús Usón, 10071 Cáceres, Spain; 6Hospital San Pedro de Alcántara, 10003 Cáceres, Spain; 7Departamento Ingeniería Electrónica, Universidad Politécnica de Madrid and CIBER-BBN, 28040 Madrid, Spain; 8Immunology Unit, University of Extremadura, 10003 Cáceres, Spain

**Keywords:** stem cells, ventricular arrhythmias, post-infarction scar, ventricular tachycardia, cardiosphere-derived cells, electrophysiology

## Abstract

Clinical data suggest that cardiosphere-derived cells (CDCs) could modify post-infarction scar and ventricular remodeling and reduce the incidence of ventricular tachycardia (VT). This paper assesses the effect of CDCs on VT substrate in a pig model of postinfarction monomorphic VT. We studied the effect of CDCs on the electrophysiological properties and histological structure of dense scar and heterogeneous tissue (HT). Optical mapping and histological evaluation were performed 16 weeks after the induction of a myocardial infarction by transient occlusion of the left anterior descending (LAD) artery in 21 pigs. Four weeks after LAD occlusion, pigs were randomized to receive intracoronary plus trans-myocardial CDCs (IC+TM group, n: 10) or to a control group. Optical mapping (OM) showed an action potential duration (APD) gradient between HT and normal tissue in both groups. CDCs increased conduction velocity (53 ± 5 vs. 45 ± 6 cm/s, *p* < 0.01), prolonged APD (280 ± 30 ms vs. 220 ± 40 ms, *p* < 0.01) and decreased APD dispersion in the HT. During OM, a VT was induced in one and seven of the IC+TM and control hearts (*p* = 0.03), respectively; five of these VTs had their critical isthmus located in intra-scar HT found adjacent to the coronary arteries. Histological evaluation of HT revealed less fibrosis (*p* < 0.01), lower density of myofibroblasts (*p* = 0.001), and higher density of connexin-43 in the IC+TM group. Scar and left ventricular volumes did not show differences between groups. Allogeneic CDCs early after myocardial infarction can modify the structure and electrophysiology of post-infarction scar. These findings pave the way for novel therapeutic properties of CDCs.

## 1. Introduction

The risk of sudden death is increased in the first months after acute myocardial infarction [1], and implantable cardioverter-defibrillator may not reduce total mortality at this time point as the rate of non-sudden deaths offsets that of sudden deaths [2,3]. The deleterious effect of shocks on the failing myocardium may partially explain these findings [4]. Therefore, new strategies to prevent the formation of ventricular tachycardia (VT) substrate early after myocardial infarction are needed.

Most post-infarction VTs have their origin in a reentrant circuit in which slow conduction pathways of heterogeneous tissue (HT) are embedded in an unexcitable dense scar. This substrate can be eliminated by catheter ablation but can also be inactivated by biological therapies. Modification of the electrophysiological properties of the HT, including prolongation of ventricular refractoriness and improvement in conduction velocity by gene transfer therapy to modify potassium repolarization currents or connexin-43 expression, have managed to reduce VT inducibility in postinfarction VT models [5,6]. Recent clinical data suggest that stem cell therapy, specifically cardiosphere-derived cells (CDCs), modifies the remodeling of the post-infarction scar. The CADUCEUS and ALLSTAR studies showed that CDCs reduced post-infarction scar size and left ventricular volumes in comparison with placebo [7,8]. These effects could explain the lower incidence of VT in treated patients, which was observed in a meta-analysis that included all randomized studies in which bone marrow-derived stem cells were used to treat post-infarction patients [9]. Nevertheless, no data on the electrophysiological effects of stem cells on VT substrate are reported. Therefore, we do not know whether the lower incidence of VT is due to a reduction in left ventricular volume or to a direct effect on the electrophysiological properties of the arrhythmic substrate.

We hypothesized that, due to its anti-fibrotic and protective effects, stem cell therapy could modify the structure and electrophysiological properties of dense scar and HT, thus preventing the development of VT substrate.

The study presented is a subanalysis of a study evaluating different routes of CDC administration on left ventricular remodeling. Specifically, this subanalysis compares the histological and electrophysiological properties of dense scar and heterogeneous tissue between a control group and a group that received CDCs via intracoronary and intramyocardial routes.

An electrophysiological study including endocardial electroanatomical mapping, in addition to epicardial optical mapping and histological analysis of the dense scar and HT, was undertaken to evaluate the effects of CDCs on the arrhythmogenic substrate.

## 2. Results

### 2.1. Scar and Left Ventricle Volume Changes

Changes in scar (total scar, dense scar, HT) and left ventricle (LV) volumes between weeks 4 and 16 are shown in Table 1. Total scar, dense scar and HT volumes increased in both groups, and no differences were found at 16 weeks. Dense scar increase was larger in the control group than in the Intracoronary–Transmyocardial (IC+TM) group (77 ± 70% vs. 31 ± 12%, *p* = 0.05). The expansion of dense scars in both groups seems to follow different mechanisms according to the relation of dense scar volume measured at 4 and 16 weeks (Figure 1). Left ventricular volumes and ejection fraction were not different between groups.

### 2.2. Electrophysiological Study and Electro-Anatomical Mapping

VT inducibility was significantly higher in the control group (100%) than in the IC+TM group (40%, *p* = 0.003). No differences were observed in the size of the endocardial scar.

### 2.3. Optical Mapping

#### 2.3.1. Mapping during Epicardial Pacing

No differences in action potential (AP) amplitude and resting potential were observed between the HT and normal zone (NZ) of myocardium. The APD was longer in the HT than at the NZ in both the control (195 ± 43 vs. 168 ± 36 ms *p* = 0.001) and the IC+TM group (273 ± 60 vs. 241 ± 53 ms, *p* = 0.01). This resulted in a gradient of APs in the scar border zone (Figure 2).

A comparison between the control and IC+TM groups is shown in Figure 3. In the IC+TM group, APD was longer and conduction velocity (CV) was faster in all areas. The slowest CV was recorded in the intra-scar channels (ISC), but there were no differences between groups (31 ± 7 vs. 30 ± 6 cm/s). The wavelength (V×APD) was shorter in the control group both in the HT (9.5 ± 2.5 vs. 15.2 ± 2.8 cm, *p* = 0.0005) and in the epicardial conduction channels (7.8 ± 2.8 vs. 10 ± 1.4 cm, *p* = 0.05).

Depolarization and repolarization isochrone maps of HT and NZ showed more dispersion of depolarization and repolarization times in the control group (central illustration). The repolarization time/depolarization time ratio was significantly larger in the control group, suggesting that CDCs have a more pronounced effect on repolarization than depolarization time.

#### 2.3.2. Mapping during VT

A VT was induced in one IC+TM and seven control hearts (*p* = 0.03). In the control groups, the entire VT circuit was in the epicardium. In five VTs, the central isthmus was an epicardial ISC. The ISC were corridors of surviving myocardium sheathing epicardial coronary artery and surrounded by scar tissue (Figure 4, and Appendix A). The remaining VTs were macroreentries around dense scar islets. In the IC+TM group, only one VT was induced and the activation sequence suggested an intramural or endocardial origin.

### 2.4. Histological Analysis

#### 2.4.1. Dense Scar

The most notable difference between the control group and the IC+TM group was the density of myofibroblasts. In the control group, myofibroblasts were observed in 60.8 ± 22% of squares vs. 35.3 ± 27% in the IC+TM group (*p* = 0.001); see Figure 5. This observation is relevant as dense scar growing was related to the density of myofibroblasts: dense scar volume increased 81 ± 66% in those animals showing myofibroblast in more than 50% of squares vs. 22 ± 16% in the remaining animals (*p* = 0.01).

#### 2.4.2. Heterogeneous Tissue

The average area per slice covered by HT was significantly larger in the control group than in the IC+TM group (36 ± 18 vs. 24 ± 9 mm2; *p* = 0.001). The area covered by fibrotic strands was also larger in the control group (19.5 ± 12 vs. 12.2 ± 7 mm^2^; *p* = 0.01). The fibrotic strands of the HT showed significant differences between the two groups. Firstly, compact or very compact collagen occupied 77% of squares of fibrotic tissue in the control group vs. 51% in the IC+TM group (*p* = 0.01). Secondly, myofibroblasts were observed in six out of eight control animals (Figure 6), but were not observed in any of the pigs receiving CDCs (*p* = 0.001). No differences in inflammatory response cells were observed between groups.

Myocyte viability was similar, as signs of degeneration were observed in 30% of myocytes in both groups. CDCs seem to improve myocyte connectivity, as 82 ± 9% of myocytes were electrically coupled to other myocytes through connexin-43-containing gap junctions in the intercalated discs in comparison to 72 ± 13% (*p* = 0.006) in the control group (Figure 7).

#### 2.4.3. Subendocardium

The average area per slice of surviving subendocardium beneath the dense scar was significantly larger in the IC+TM CDC group than in the control group (27 ± 14 vs. 16 ± 8 mm^2^, *p* = 0.01). As observed in other areas, myofibroblasts were not present in any sample from the IC+TM group but they were identified in three control samples (*p* = 0.01). Whereas in the control group only a thin layer of almost normal myocytes survived, in the IC+TM group these layers were thicker but contained more degenerated myocytes. This explains why, in the control group, the subendocardium had fewer myocytes with signs of degeneration (25% vs. 63%, *p* = 0.020) but better distribution of connexin-43 (74% vs. 64%, *p* = 0.040).

#### 2.4.4. Epicardial Intra-Scar Channels

Myocyte degeneration seems to be less frequent in the IC+TM CDC group, in which only 27% of myocytes showed signs of degeneration vs. 40% in the control group (*p* = 0.10). Connexins 43 were normally distributed in 40 and 31% of myocytes in the IC+TM CDC and control groups (*p* = 0.20), respectively. Myofibroblasts were only identified in two ISCs of control pigs.

## 3. Discussion

This study describes for the first time the electrophysiological and structural remodeling, as well as the effect of CDCs, in a model of postinfarction VT.

### 3.1. Structural and Electrophysiological Remodeling in Healed Infarct

In both groups, scar, dense scar and HT volumes increased between the first and second magnetic resonance image (MRI). This implies that the scar is a dynamic entity where the dense scar expands at the expense of HT and HT expands at the expense of surrounding normal tissue. If the transformation of HT into a dense scar were the only phenomenon taking place, HT volume at 16 weeks would have diminished. The expansion of dense scars could be explained by two mechanisms:
The dense scar replaces myocyte loss at the border zone. In this case, the primary event is myocyte death.Dense scar growth is the primary event, thus dense scar surrounds and isolates myocytes, expediting cellular death somehow.

As scar growth was related to myofibroblast density in the dense scar, it seems that the higher the density of myofibroblasts, the greater the capacity to infiltrate the surrounding tissue. This fact could explain scar growth differences between groups.

The optical mapping provided relevant information on the electrophysiological characteristics of the healed infarct. First, the APD was longer in ISCs and HT than in NZ. The fact that the longest APD was recorded in ISCs could favor VT inducibility. Short-coupled extra-stimuli that cross the HT reach the ISC when they are still depolarized, and thus can set the stage for a unidirectional block at the entrance of ISCs, allowing the establishment of a stable re-entry. We do not have a clear explanation for APD gradient, but a lower sink effect due to borders in the unexcitable dense scar and heterogeneous distribution of delayed rectifier K-currents in HT could be involved [10]. Conduction velocity that was significantly slower in ISC and HT than at NZ could also favor VT inducibility. Slow conduction velocity could be explained by the lateralization of connexin-43 and fibrosis infiltration, given that no differences in AP amplitude or resting potential were observed.

The isthmuses of most VT were found in ISCs; these channels cross the dense scar and create the substrate for reentry. ISCs are corridors of myocytes that survive because of their proximity to epicardial coronary arteries. All VTs induced in the control group were located in the epicardium, probably related to the pacing site being in the epicardium and closer to the epicardial circuits and not due to the lack of other circuits in other locations.

### 3.2. Effects of CDCs on Structural and Electrophysiological Remodeling in Healed Infarct

This study showed that CDCs did not eliminate pre-existing scars but modified the histology of the dense scar, and HT modified the mechanism of scar expansion as suggested by the relation of dense scar volumes between weeks 4 and 16 (Figure 1). CDCs could delay scar growth by blocking the infiltration of the surrounding tissue by dense scar. In this sense, the mobilization of stem cells in the infarct area can protect and prevent the advance of tissue deterioration [11]. Myofibroblasts were significantly more numerous in the control group than in the IC+TM CDC group. Myofibroblasts in the dense scar could favor the progression of fibrosis and increase collagen density and CDCs which may delay the penetration of fibrosis into normal tissue by blocking the trans-differentiation of fibroblasts into myofibroblasts [12]. In this sense, and despite the inflammatory effect possibly reducing inflammatory response [13], we did not observe significant inflammatory differences between groups. Myofibroblasts were also found in the HT of control animals, but were not in contact with myocytes as they tend to be in the central parts of the fibrotic strands. Therefore, the electrophysiological interaction between myocyte/myofibroblast seems highly improbable [14].

Optical mapping also showed that CDCs prolong APD and increase CV, resulting in a longer wavelength, and therefore larger circuits are required to support stable reentries. Moreover, CDCs reduced the repolarization time/depolarization time ratio, suggesting a shortening of repolarization beyond the effect on depolarization. A short repolarization time implies less APD dispersion during repolarization, and consequently reduces the time during which a myocyte can re-excite other myocytes. Several mechanisms, such as improvements in myocyte connectivity, the reduction of interstitial fibrosis, and the generation of novel cardiomyocytes [15], could explain CDC-induced changes. The results from our histological analyses (reduction of fibrosis at the border zone and better myocyte connectivity) support these mechanisms.

### 3.3. Safety Concerns of Stem Cell Therapy

Previous reports concerning the effects of CDCs on VT inducibility are sparse and not conclusive [16]. None of the animals included in the study experienced sudden death following cell implantation. Only skeletal myoblasts and mesenchymal stem cells have been associated with an increase in ventricular arrhythmias in the early postimplantation period [17]. Proarrhythmic effects have not been described with CDCs [7,8] or cardiac stem cells [18].

### 3.4. Limitations

The results of this study, including the differences in VT inducibility, should be taken with caution due to the limited number of animals included. Optical mapping does not explore the endocardium, so electrophysiological data at this level are lacking. Ventricular stimulation was only performed from one site, therefore we do not have information concerning the effect of the direction of the activation wavefront.

## 4. Materials and Methods

All experiments with live animals were approved by the Institutional Animal Care and Use Committee (Centro de Cirugía de Mínima Invasión Jesús Usón). Animals were obtained from the animal housing facility (Centro de Cirugía de Mínima Invasión Jesús Usón), which has been certificated to produce and investigate laboratory animals in compliance with Spanish and European Legislation. A detailed Methods section is provided in Appendix B.

### 4.1. Ventricular Tachycardia Substrate Evaluation Sub-Study

This sub-study was part of a study evaluating the effect of different CDC injection routes on scar remodeling and ventricular function, and was conducted to determine the effects of CDCs on the electrophysiology and structure of HT and dense scar of healed infarcts (study and sub-study protocols are shown in Figure 8). A porcine model characterized by large heterogeneous scars with high VT inducibility rates was used [19]. To induce closed-chest myocardial infarction, the left anterior descending coronary artery was occluded transiently by a balloon catheter placed immediately distal to the first diagonal branch for 150 min followed by reperfusion (Appendix B). A ce-MRI study was performed at weeks 4 and 16 to determine the evolution of scar and ventricular volumes and function (Appendix B). After the first ce-MRI, study animals were randomized. The sub-study included 11 pigs randomized to the control group and 10 pigs randomized to receive intracoronary and trans-myocardial CDC (IC+TM group). After the second ce-MRI, endocardial electroanatomical mapping was performed. The heart was then explanted for optical mapping and histological evaluation of the arrhythmic substrate.

### 4.2. Cell Isolation and Production

CDCs were produced in the Laboratory of Cell Biology at the CCMI. After the first ce-MRI, CDCs were delivered using an intracoronary and trans-myocardial approach.

#### 4.2.1. Isolation and Production of Porcine Cardiosphere-Derived Cells

CDCs were obtained from cardiac tissue explants of euthanized Large White pigs. Auricular explants (1–2 g) were washed and mechanically disrupted into 1–2 mm^3^ fragments. These fragments were washed again to eliminate cellular debris. The tissue was then subjected to three successive enzymatic digestions with a solution of 0.2% trypsin and 0.2% collagenase IV at 37 °C for 5 min each. The digested tissue was washed and cultured with Complete Explant Medium (10% fetal bovine serum, 1% penicillin-streptomycin, 2 mM L-glutamine, and 0.2 mM 2-mercaptoethanol in IMDM) at 37 °C and 5% CO_2_.

After 3 weeks, tissue fragments were discarded and fibroblasts-like cells migrating from tissue explants were trypsinized and seeded in poly-D-lysine coated plates with Cardiosphere Growing Medium (10% fetal bovine serum, 1% penicillin-streptomycin, 2 mM glutamine, and 0.1 mM 2-mercaptoethanol in 35% IMDM and 65% DMEM-Ham’s F12). Under these conditions, suspended cell clusters called cardiospheres are formed, and cells migrating from them are the CDC. These cells were selected and expanded at 37 °C and 5% CO_2_. CDCs in passages 5 to 10 were used for the subsequent studies.

#### 4.2.2. Characterization of Cardiosphere-Derived Cells

The complete characterization of CDCs included a phenotypic analysis by flow cytometry, a molecular analysis of the expression of relevant markers by RT-PCR, and an analysis of the differentiation potential of CDCs toward adipogenic, chondrogenic, and osteogenic lineages.

For flow cytometric analysis, the cells were stained with FITC-conjugated monoclonal antibodies against human CD90 (porcine crossreactive), and FITC-conjugated porcine monoclonal antibodies against CD29, CD31, CD44, CD45, CD61, CD105, CD117, Sca-1, SLA-I (Swine Leukocyte Antigen class I) and SLA-II (Swine Leukocyte Antigen class II) from Serotec. The phenotypic analysis was performed as follows: = 2 × 105 cells were incubated for 30 min at 4 °C with appropriate concentrations of monoclonal antibodies. The cells were washed and resuspended in PBS. The flow cytometric analysis was performed on a FACScalibur cytometer (BD Biosciences, Franklin Lakes, NJ, USA) after the acquisition of 105 events. Cells were primarily selected using forward and side scatter characteristics and fluorescence was analyzed using CellQuest software (BD Biosciences). Isotype-matched negative control antibodies were used in all the experiments. The mean relative fluorescence intensity was calculated by dividing the mean fluorescent intensity (MFI) by the MFI of its negative control.

To analyze the expression of different markers by RT-PCR, the total RNA from CDCSs was isolated. The cDNA was synthesized from 1 μg of RNA in a reverse transcription reaction for 1 h at 37 °C using Superscript III reverse transcriptase (Invitrogen, Thermo Fisher Scientific, Waltham, MA, USA) and appropriate primers designed for Sus scrofa. Conventional PCR amplification was performed using the Taq DNA Polymerase Recombinant kit (Invitrogen) in a PXE 0.2 thermocycler (Thermo Fisher Scientific, Waltham, MA, USA). Gene expression levels were analyzed and normalized with the GeneTools software (Syngene, Synoptics group, Cambridge, UK) using beta-actin as a housekeeping gene. The relative quantification was made by measuring the brightness intensity of each band using the GeneSnap software (Syngene, Synoptics group, Cambridge, UK).

Finally, the differentiation assay of CDCSs was performed when the cells reached 80% of confluence. Cells were maintained for 21 days with a standard medium (control) or with specific differentiation media for adipogenic, chondrogenic, and osteogenic lineages. Differentiation was evidenced by optical microscopy using specific stainings = Oil Red O for adipocytes, Alcian Blue for chondrocytes, and Alizarin Red S for osteocytes.

### 4.3. Intracoronary and Transmyocardial Cell Delivery

Coronary artery cannulation was performed transfemorally (see Appendix B). Under sterile conditions, right femoral arterial access was obtained using the modified Seldinger technique, and a 7 Fr vascular sheath (Terumo, Inc., Tokyo, Japan) was placed in the femoral artery. Under fluoroscopic guidance (Philips Mobile Digital Angiographic System-BV Pulsera, Philips Medical Systems, Best, The Netherlands) a 6 Fr hockey stick guiding catheter (Mach 1 ^®^, Boston Scientific Corporation, Natick, MA, USA) was introduced and placed at the origin of the left coronary artery. For intracoronary (IC) delivery, a coaxial 21F microcatheter (Micro Ferret Infusion Catheter, Cook Medical, Bloomington, IN, USA) was placed upstream of the left anterior descending artery occlusion, and CDCs were injected at a flow rate of 1 mL/minute (total dose 300.000 CDCs/kg). A coronary angiogram to assess coronary flow was obtained 5 min after CDC injection. The arterial sheath was then removed and hemostasis was obtained via manual compression of the puncture site for at least 10 min.

For TM CDCs, the NOGA XP Cardiac Navigation System (NOGA^®^ XP Cardiac Navigation System, Biosense Webster, Diamond Bar, CA, USA) was used by an investigator with certified training. Using the NOGASTAR catheter, a three-dimensional electroanatomical map of the left ventricular endocardium was created, the border zone was identified on voltage maps, and the areas with multicomponent and LP were marked for CDC injection with the Myostar catheter. A total of 3 million CDCs were injected at 5 different sites (0.2 mL per injection).

### 4.4. Electrophysiological Study and Electro-Anatomical Mapping

Animals underwent an electrophysiological study 2 days after the second ce-MRI. This procedure has been previously described [19]. After voltage maps were obtained, programmed electrical stimulation from the right and left ventricles was performed at 3 different cycle lengths with up to 4 extra stimuli [20]. Both sustained VT and VF induced during PES were included in the analysis. Operators were blinded to the treatment group and ce-MRI images.

### 4.5. Optical Mapping

Optical maps were obtained during continuous pacing and induced VT. Briefly, CV and action potential duration (APD) were measured during epicardial right ventricle pacing. The pacing protocol consisted of the application of a biphasic pulse of 10 V at 2 ms duration with a cycle length of 800 ms from the right ventricle epicardium so that the propagation was longitudinal to epicardial fiber orientation. APD of optical voltage signals was calculated at 80% repolarization. CV was automatically estimated by calculating gradients of normal vector fields on isochronal propagation lines [21]. CV and APD were measured in the area covered by the camera where three areas were differentiated: (1) the NZ, (2) the HT surrounding the dense scar, characterized by the presence of muscle fibers and fibrotic strands and (3) the HT forming ISC. Side-by-side comparison of the epicardial image and the optical map served to differentiate NZ, HT, and ISC, and the absence of fibrotic strands upon visual inspection of the area covered by the camera served as the basis to differentiate the NZ. Once the 3 previously described zones were outlined, a mean value of APD and CV was obtained for each one of them on depolarization maps. The wavelength was calculated as CV × APD for both the border zone and the conduction channels. Activation and repolarization maps were obtained to calculate the depolarization and repolarization times in the area that included the NZ and HT. VT inducibility was tested with 10 pulse bursts followed by extra-stimuli with decreasing coupling intervals. Once a VT was mapped and terminated, the induction protocol was repeated to induce a different VT, but either the previous VT or a polymorphic VT was induced.

### 4.6. Histological Analysis

The heart was cut into transverse slices from apex to base. Two to five slices (depending on scar size) covering the whole scar were analyzed per animal. Each previous slice was cut into new 4-μm thick slices which were used for Masson’s Trichrome stain and immunohistochemistry with protein and cell-specific antibodies. With Masson’s Trichrome stain, three types of tissue were differentiated: 1. dense scar, defined by the absence of myocytes; 2. the HT, characterized by the presence of myocytes and strands of fibrosis that separate or isolate the myocyte bundles; and 3. the NZ, defined by normal myocytes and the absence of fibrosis. The infarcted area was differentiated into 4 substructures, which were analyzed separately (Figure 9): 1. central dense scar; 2. the subendocardium, the tissue spanning from the endocardium layer to the overlying dense scar; 3. The lateral HT, including the HT surrounding the dense scar excluding the subendocardial; and 4. epicardial intra-scar channels (ISC).

The histological analysis of dense scar included the semiquantitative characteristics of collagen fibers, the density of vessels, and the type and density of different cells. The subendocardium, lateral HT, and the ISC analysis comprised the measurement of the area in the transverse slide, the area covered by fibrosis, and within the fibrosis the density of vessels and the type and density of different cells. In myocytes, we analyzed viability and connectivity. Myocyte viability was defined by the absence of hypertrophy, myocytolysis, edema, and vacuolar degeneration. To define myocyte connectivity, a specific marker for connexin-43 (Cx43) was used and two variables were measured: 1. percentage of myocytes in the HT with a normal distribution of connexins, and 2. percentage of myocytes in the HT that were disconnected from other myocytes. For the identification and quantification of the different types of cells (myocytes, fibroblasts, myofibroblasts, vascular and non-vascular smooth muscle cells (SMC), vascular endothelial cells, proliferation, and inflammatory response cells) we used cell-type-specific markers for myosin heavy chain 6 (α-MYH6), myosin heavy chain 7 (β-MYH7), myosin light chain (MLC), α-smooth muscle actin (α-SMA), CD31, Von Willebrand factor (vWF), Ki67, CD3, and CD20.

Measurements were made with a transparent grid of 20 × 40 mm, divided into 1250 squares (0.64 mm^2^ per square). The presence of collagen was determined by the Masson’s Trichrome stained sections under polarized light. Dense scar, subendocardium, lateral HT, and epicardial ISC areas were measured by counting the number of squares and multiplying by 0.64 mm^2^. To compare myocyte viability and connectivity, all myocytes in the grid (20 × 40 mm) were counted and the percentage of normally preserved myocytes (absence of vacuoles, edema, myocytolysis) and the percentage of myocytes with a normal distribution of connexin-43 along their longitudinal axis, were determined. The spreading of different types of cells (fibroblasts, myofibroblasts, vascular and non-vascular smooth muscle cells, endothelial cells, and inflammatory response cells) was quantified by calculating the percentage of squares showing a particular cell. The investigators were blinded to the treatment group.

### 4.7. Statistical Analysis

Continuous variables are presented as mean ± SD, and categorical data are summarized as frequencies and percentages. Statistical significance of the differences between groups was assessed using Student’s *t*-test for normally distributed data or Wilcoxon’s test for non-normally distributed variables. Categorical data were compared using Chi-squared and Fisher’s tests. A 2-sided *p* value of <0.05 was considered to indicate statistical significance. Statistical analysis was performed using the JMP statistical software package (JMP Inc., Cary, NC, USA).

## 5. Conclusions

Allogeneic CDCs early after myocardial infarction modify the structure and electrophysiology of post-infarction scar. These findings suggest novel therapeutic actions of CDCs.

## Figures and Tables

**Figure 1 ijms-23-16211-f001:**
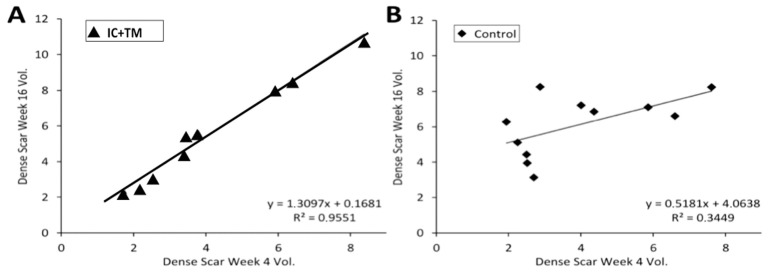
Regression line showing the correlation of dense scar volume between week 4 and week 16. (**A**): Dense scar correlation in the IC+TM CDCs group. (**B**): Dense scar correlation in the control group. The regression line shows a strong association between dense scars at weeks 4 and 16 in IC+TM group, suggesting scar growth is similar for all sizes of scars. In the control group, the smallest scars grew the most. IC: Intracoronary; TM: Transmyocardial; CDC, Cardiosphere-derived cells.

**Figure 2 ijms-23-16211-f002:**
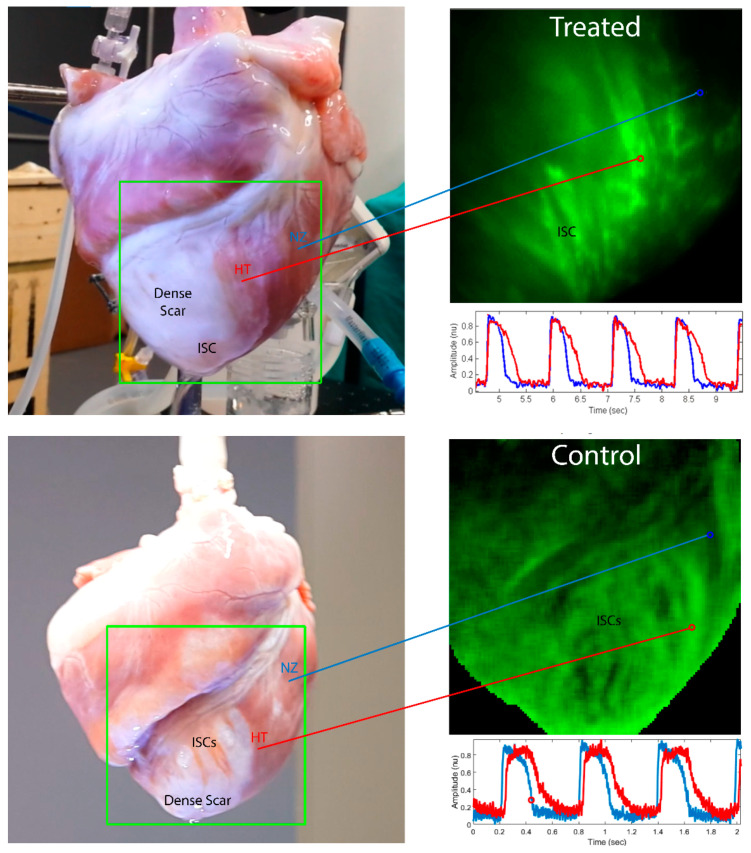
Upper panel: Optical mapping study showing the APD gradient around the scar. Blue lines represent AP in NZ areas and red lines HT APs. (**Upper panel**): Epicardial surface of a heart from the IC+TM treated group. The area that was optically mapped (shown inside the green rectangle) includes dense scar (absence of muscle), HT (fibrotic strands separating myocytes bundles), and NZ where no fibrosis is seen. Optical mapping of the area delimited by the green frame shows the APD gradient between HT and NZ; the HT remains green (myocytes are still depolarized), whereas the NZ has turned black (myocytes have repolarized). Below the optical mapping image, the AP of HT and NZ are shown. APs from HT were longer but no differences in amplitude and resting membrane potential were observed. (**Lower panel**): Optical mapping showing the APD gradient around the scar. On the left, the epicardial surface of a control heart is presented with the area that was optically mapped and delimited by a green rectangle. The figure shows the dense scar (absence of muscle) with intra-scar channels running along epicardial arteries, the HT, characterized by fibrotic strands separating myocytes bundles (red arrow marks the limits), and the NZ where no fibrosis is seen. The optical mapping of the area delimited in green shows the APD gradient between HT and NZ; the HT remains green (myocytes are still depolarized), whereas the NZ has turned black (myocytes have repolarized). In this example, the HT is activated with some delay with respect to the NZ, as is shown in the lower corner of the panel by the simultaneous recording of HT and NZ AP. No differences in resting membrane potential were observed, but AP amplitude was slightly lower in the HT. APD, action potential duration; HT, heterogeneous tissue; NZ, normal zone.

**Figure 3 ijms-23-16211-f003:**
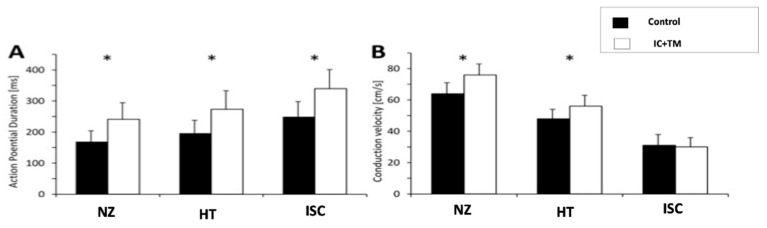
Differences between control and IC+TM groups on action potential duration (**A**) and conduction velocity (**B**) during optical mapping. (*) Statistically significant *p* value < 0.05. ISC, epicardial intra-scar channel; HT, heterogeneous tissue; IC, intracoronary; TM, transmyocardial; NZ, normal zone.

**Figure 4 ijms-23-16211-f004:**
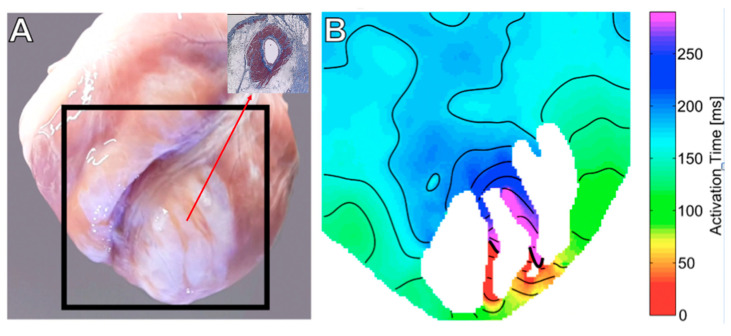
(**A**): Example of a control experiment showing two ISCs of surviving myocardium; the red arrow shows the cross-section of one ISC, revealing viable muscle around the epicardial artery. These channels course along epicardial arteries. These channels served as the central isthmus of an induced VT whose isochron map is shown in (**B**) (See also Appendix A). ISCs, intra-scar channels; VT, ventricular tachycardia.

**Figure 5 ijms-23-16211-f005:**
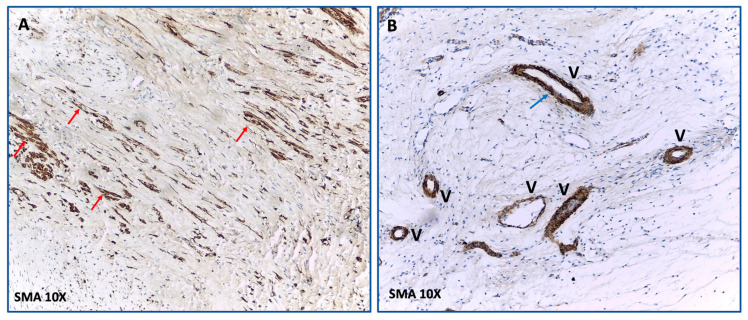
This figure shows the histological differences in dense scars between the control and IC+TM groups. (**A**): The control group is characterized by the presence of numerous myofibroblasts (brown elongated cells; red arrows) that stain with anti-α-SMA antibodies and that are not related to vessels. (**B**): In the IC+TM experiment, the anti-α-SMA antibodies only stain the walls of vessels (V) where smooth muscle is present (blue arrow). α-SMA: α-smooth muscle actin.

**Figure 6 ijms-23-16211-f006:**
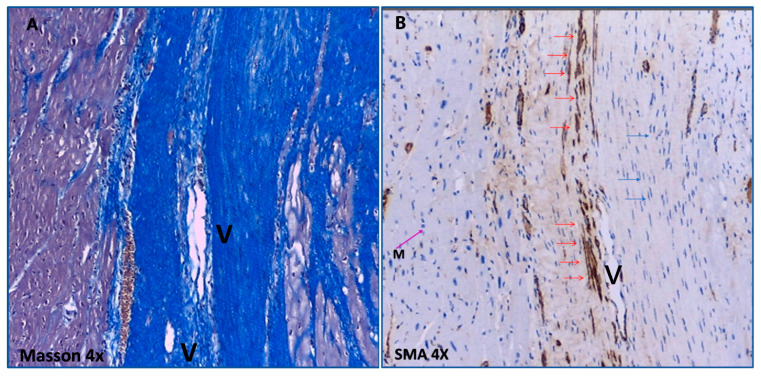
Characteristics of the fibrotic tissue at the HT in a control experiment. (**A**): Masson’s Trichrome stains blue the collagen tracts that invade and separate the bundles of myocytes at the border zone. (**B**): In the same area, anti-α-SMA antibodies reveal the presence of myofibroblasts (brown elongated cells, red arrows) in the collagen tracts separating bundles of myocytes. α-SMA: α-smooth muscle actin.

**Figure 7 ijms-23-16211-f007:**
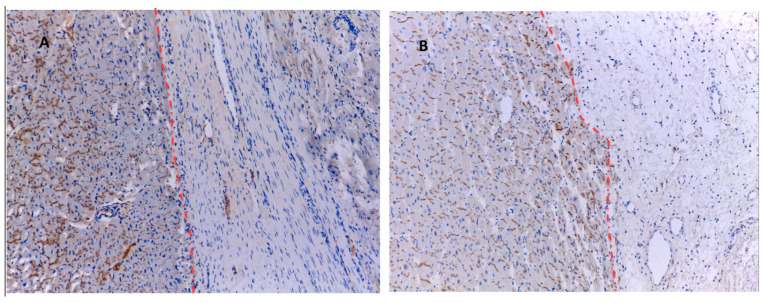
Distribution of connexin-43 in HT. The dashed red line marks the boundaries between fibrotic tissue and myocyte bundles. (**A**): connexin-43 (brown bands) distribution in the control group is more irregular than in the IC+TM group (**B**), in which most myocytes at the border zone were connected to other myocytes by connexin-43 located in intercalated discs.

**Figure 8 ijms-23-16211-f008:**
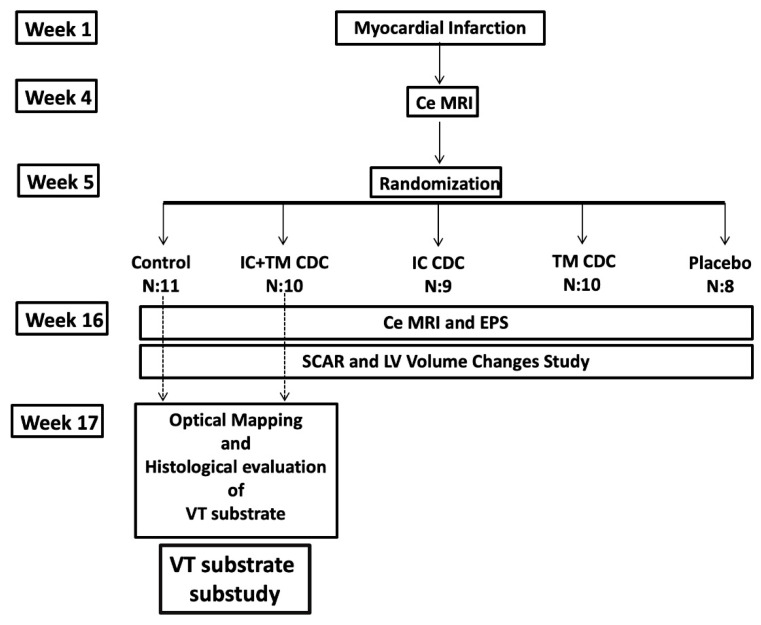
Flow diagram showing the VT substrate substudy protocol within the protocol that analyzed the effects of CDC on the scar and LV volumes with ceMRI. CDCs: Cardiosphere-derived cells, ce-MRI: contrast-enhanced magnetic resonance imaging, EPS: Electrophysiological Study, IC: Intracoronary, TM: Transmyocardial.

**Figure 9 ijms-23-16211-f009:**
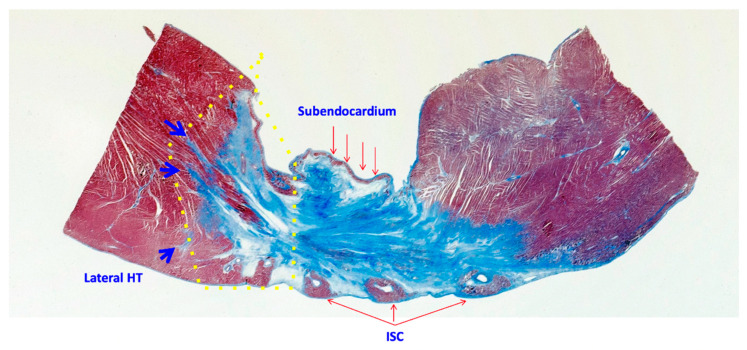
Masson’s Trichrome stain shows a cross-section of the scar with: 1. Central dense scar (blue area devoid of myocytes), 2. Subendocardium, 3. The lateral HT (the area inside the yellow dotted line) in which the external boundary was marked by the tip of the fibrotic strands (blue arrows) and 4. The intra-scar channels. HT: heterogeneous tissue; EISC, epicardial intra-scar channels.

**Table 1 ijms-23-16211-t001:** Scar and ventricular volume changes between weeks 4 and 16. HT, heterogeneous tissue; IC, intracoronary; TM, transmyocardial; LVEF, left ventricular ejection fraction; ns, not significant.

	Control (N = 11)	IC+IM Cells (N = 9)	*p*-Value
Scar 4W (cc)	7.4 ± 3.2	8.2 ± 3.0	ns
Scar 16W (cc)	11.0 ± 4.0	11.2 ± 4.0	ns
∆Scar (%)	80.0 ± 80.0	36.0 ± 17.0	0.1
Dense Scar 4W (cc)	3.9 ± 2.0	4.1 ± 2.2	ns
Dense Scar 16W (cc)	6.1 ± 1.7	5.5 ± 3.0	ns
∆Dense Scar (%)	77.0 ± 70.0	31.0 ± 12.0	0.02
HT 4W (cc)	3.5 ± 1.5	4.0 ± 1.1	ns
HT 16W (cc)	5.8 ± 2.7	5.6 ± 1.6	ns
∆HT (%)	64.0 ± 69.0	42.0 ± 31.0	ns
LVEF % (4W)	44.0 ± 6.0	40.0 ± 6.0	0.1
LVEF % (16W)	39.0 ± 5.0	40.0 ± 6.0	0.9
∆LVEF (%)	−10 ± 14.0	1.0 ± 8.0	0.04
Scar 4W (cc)	7.4 ± 3.2	8.2 ± 3.0	ns

## Data Availability

The data in this study are available from the corresponding author on request.

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
