# Peer review of "Effects of Cardiac Stem Cell on Postinfarction Arrhythmogenic Substrate"

_ijms, 2022, doi:10.3390/ijms232416211_

Round 1

Reviewer 1 Report

Excellent work, congratulations for the authors.

Author Response

We greatly appreciate the comment and congratulations received.

Reviewer 2 Report

The paper is written well. It does not need any language editing. Reference list is adequate. I don’t have any comments on tables and figures. The topic is of interesting since there are evidences on stem cell therapy in postischemic cardiomyopathy and the issue of heterogeneous conduction velocity leading to cardiac arrhythmia (VT) is of interest.

Author Response

We strongly appreciate the time spent by the reviewer and the positive feedback regarding our work.

Reviewer 3 Report

Please re-write in a more intellegible way the Introduction, focusing on what is known, gaps, hypothesis and study aim.

Please describe more in detail, in the Methods:

-the technique to isolate and produce CDCs;

-the cardiac injective technique of CDCs. According to literature, did you use intra-myocardial injection or intra-coronary injection? Please fully describe this point.

-did you use anti-inflammatory or immunosuppressive techniques for CDCs implantation? Pelase describe it.

-I do not understand a point. Did you remap or not the ventricle of patients after the CDCs (vs. controls) injection or not? Did it modify the scar area and inducible protocols? Please describe this point. If the answer is not, it is a study limitation.

-In a previous article, (Int J Cardiol. 2013 Oct 9;168(4):3954-62. doi: 10.1016/j.ijcard.2013.06.053), authors showed that a peri-procedural tight glycemic control during early percutaneous coronary intervention up-regulated endothelial progenitor cell level and differentiation during acute ST-elevation myocardial infarction. This effects was relevant to reduce scar area extension, and induce best clinical outcomes (Int J Cardiol. 2013 Oct 9;168(4):3954-62. doi: 10.1016/j.ijcard.2013.06.053). do you have data on stem cells mobilization during VT ablation or not? Please discuss this point.

-improve English quality of the text.

Author Response

Firstly, we would like to express our sincere gratitude to the reviewer for the time and effort made in the revision of our manuscript. The recommendations that have been put forward surely will help improve the quality of our work.

  • As requested by the reviewer, the introduction section has been rewritten following the points proposed by the reviewer for a better understanding of the manuscript.
  • Following the comments of the reviewer we have modified the Methods section:
    • Regarding the technique employed to isolate and produce the CDCs, we have moved the description in Appendix A into the Methods section of the new version of the manuscript.
    • With respect to the cardiac injection procedure of CDCs, we used both intracoronary and intracardiac injections. We used the term Transmyocardial (TM) for the latter. The cells were injected employing a NogaStar® Catheter into the areas with late potentials, as detailed in the last paragraph of the Methods section “5.3 Intracoronary and transmyocardial cell delivery”.
    • In respect of anti-inflammatory or immunosuppressive therapies, we did not use anti-inflammatory or immunosuppressive therapies because Allogeneic CDC transplantation without immunosuppression is safe. (Circulation. 2012;125:100–112).
    • As pointed out by the reviewer with reference to the remapping and inducibility protocols, we explained in the Methods section “5.1. Ventricular Tachycardia Substrate Evaluation Sub-study”, we only mapped the left ventricle after the second ceMRI: “After the second ce-MRI an electrophysiological study including endocardial electroanatomical mapping was performed”. 

As the reviewer comments, comparing the inducibility of VT before randomization and at the end of the study would have provided very relevant information on the effect of the cells on inducibility and on the change in the size of the endocardial scar; however, this would imply another invasive study with possible loss of animals. On the other hand, we consider that with the 2 ceMRIs we could perfectly assess the effect of the cells on the evolution of the scar without the need to perform an endocardial map at 4 weeks. We also considered that the importance of the work presented was to study the inducibility at the end of the study and relate it to the electrophysiological parameters of HT and the histological differences in the scar.  The effect of CDCs on the evolution of the scar was assessed by comparing the two ceMRI at 4 and 16 weeks.

  • Regarding the stem cells mobilization during VT ablation, the animals did not undergo VT ablation. Therefore if the question is whether we evaluated the mobilization of the progenitor cells after CDC implantation, the answer is no.
  • As pointed out by the reviewer the English quality of the manuscript has been addressed accordingly.

Reviewer 4 Report

In the study, titled "Effects of cardiac stem cell on postinfarction arrhythmogenic substrate", Arenal A et al., describe the role of cardiosphere derived cells (CDCs) for prevention of post infarction associated development of arrhythmias. Authors show how transplantation of CDCs prolongs action potential duration and conduction velocity in the heart after infarction injury. Overall, the results are interesting but are observational and do not provide any mechanistic details. Authors should expand the introduction section to discuss studies that show a similar effect on arrhythmogenic substrate development in response to cell-based therapies and how CDC effect compares to it can be further elaborated in the discussion section. Also, it would be helpful to highlight the novelty of the findings at the methodological or observational level.

1. Acronyms and abbreviations should be described at first use.

2. Figure 2: Unclear whether the pictures and the line graph are representative image for control or IC+IM group. Please provide images and data for both groups. Is there a difference within AP in the HZ and NZ between the 2 treatment groups?

3.  Authors should note in the discussion that experiments regarding induced VT in the IC+IM group are limited and preliminary since only one samples was used.

4. Please add a bar graph in Figure 5 for SMA quantitation. Also, authors should provide some analysis of how they were able to differentiate between myofibroblasts and smooth vessels as SMA labels both.

5. Please add bar graphs for all staining quantitation in Figure 6 and 7. 

Author Response

We are grateful to the reviewer for the time spent revising our manuscript and for considering that our work is interesting. We appreciate all the comments and recommendations made and are thoroughly convinced they have enabled us to improve the quality of our work. All the recommendations made have been incorporated into the modified version of the manuscript.

  • As the reviewer rightly pointed out, we have modified the introduction and method sections to better complement the discussion section.
  • We have carefully reviewed all acronyms and abbreviations used in the manuscript and have corrected those that were not defined in their first use.
  • We agree with the comment regarding Figure 2. We have modified it to present a clearer and more representative example of the comparison between control and treatment groups. In the new version, we provide images of a IC+TM group example (upper panel) and control group (lower panel). Consequently, the caption of the figure has been updated.
  • We appreciate the comment regarding the discussion section. We have added this point to the discussion.
  • As suggested by the reviewer, we have modified Figure 5. We have embedded bar graphs representing the percentage of the staining quantitation of the examples displayed in the figure. In figure 5B the SMA is only seen in the wall of the vessels (blue arrows) and it was not accounted for the histological analysis.
  • Similarly to Figure 5, we have included the bar graphs as requested by the reviewer.

Round 2

Reviewer 3 Report

Dear authors,

I see improvement in the manuscript after the revision process. By the way, I would suggest you to fix the following points:

-please, remember the endogen and spontaneous mobilization of stem cells during acute phase of myocardial infarction, and induced by mechanisms of cellular protection and repair, as those seen post coronary intervention (Int J Cardiol. 2013 Oct 9;168(4):3954-62. doi: 10.1016/j.ijcard.2013.06.053). in this case, the pci could induce mobilization and migration of endothelial progenitor cells, to repair the acute cardiac damage and reduce the scar extension (Int J Cardiol. 2013 Oct 9;168(4):3954-62. doi: 10.1016/j.ijcard.2013.06.053). Please discuss this point.

-Again, few drugs’ therapies could induce the same effects in humans with acute myocardial infarction (Cardiovasc Diabetol. 2022 May 15;21(1):77. doi: 10.1186/s12933-022-01506-8; Biomedicines. 2021 Jul 28;9(8):904. doi: 10.3390/biomedicines9080904.), by the control of cellular ionic channels and significant reduction of inflammatory burden. What is your opinion? Please discuss it. 

Please short the Discussion paragraph, focusing on the main study results.

Author Response

We thank the reviewer for the work and recommendations that have been made to improve the manuscript.

We have revised the references and added them in the discussion section as suggested. With respect to the inflammatory response since we did not find any significant change between groups, we cannot provide any further discussion.

Once again thank you for your time and consideration reviewing the article.

Reviewer 4 Report

There are no further comments.

Author Response

We will like to thank the reviewer for the work and recommendations that allowed us to improve the manuscript quality.